# Co-Reinforcement Learning for
# Unified Multimodal Understanding and Generation

**Jingjing Jiang**[1,2]    **Chongjie Si**[1]    **Jun Luo**[2]    **Hanwang Zhang**[2]    **Chao Ma**[1,*]

[1] Shanghai Jiao Tong University    [2] Nanyang Technological University

jingjingjiang2017@gmail.com, {chongjiesi,chaoma}@sjtu.edu.cn
{junluo,hanwangzhang}@ntu.edu.sg

## Abstract

This paper presents a pioneering exploration of reinforcement learning (RL) via group relative policy optimization for unified multimodal large language models (ULMs), aimed at simultaneously reinforcing generation and understanding capabilities. Through systematic pilot studies, we uncover the significant potential of ULMs to enable the synergistic co-evolution of dual capabilities within a shared policy optimization framework. Building on this insight, we introduce **CoRL**, a **Co-R**einforcement **L**earning framework comprising a unified RL stage for joint optimization and a refined RL stage for task-specific enhancement. With the proposed CoRL, our resulting model, **ULM-R1**, achieves average improvements of 7% on three text-to-image generation datasets and 23% on nine multimodal understanding benchmarks. These results demonstrate the effectiveness of CoRL and highlight the substantial benefits of reinforcement learning in facilitating cross-task synergy and optimization for ULMs. Code is available at https://github.com/mm-vl/ULM-R1.

## 1  Introduction

As large foundation models (LFMs) continue to advance in their general capabilities and breadth of knowledge, post-training [30, 46, 70, 73, 108] has emerged as a critical paradigm for further refining pretrained LFMs toward specialized applications, thereby facilitating task adaptation and human-aligned behaviors. Recently, reinforcement learning (RL)-based approaches [51, 52, 59, 64, 69, 72, 95] have exhibited considerable promise due to their data efficiency and strong alignment abilities. A notable exemplar is DeepSeek-R1 [19], which demonstrates that RL with verifiable rewards and the group relative policy optimization (GRPO) algorithm constitutes a practical and stable strategy that sidesteps explicit preference modeling [77] and reward model learning [83]. This promising paradigm indicates *the significant potential of LFMs to acquire advanced capabilities and generalize effectively without dependence on large-scale, high-quality supervised data*.

In the multimodal AI research community, the prevailing implementation [9, 24, 40, 41, 53, 69, 100, 101] of the GRPO algorithm centers on crafting diverse rule-based reward mechanisms to incentivize long-chain reasoning capabilities of multimodal large language models (MLLMs). These initiatives primarily target multimodal understanding, with a particular focus on visual and mathematical reasoning tasks. Conversely, its application to visual generation remains surprisingly limited, with only pioneering explorations [25, 79] suggesting its feasibility. More importantly, extending GRPO to unified MLLMs (ULMs) [8, 39, 88, 90] capable of concurrently performing visual understanding and generation tasks remains considerably under-explored. Intuitively, ULMs could significantly

---

*Corresponding author.

39th Conference on Neural Information Processing Systems (NeurIPS 2025).

benefit from GRPO owing to their inherent advantages of *cross-task synergy* and *LLM sharing*, which enables ULMs to share reward signals across various tasks and effectively mitigate reward imbalance, particularly as GRPO operates by jointly ranking outputs within task-agnostic groups.

This paper aims to enhance the understanding and generation capabilities of ULMs without relying on supervised data. We begin with a set of pilot experiments to explore efficient reinforcement learning paradigms. Specifically, we systematically examine four rule-based training strategies: (*i*) separate RL for individual tasks, (*ii*) separate RL with weight merging, (*iii*) cycle RL alternating between tasks, and (*iv*) unified RL with joint optimization. Our explorations reveal two critical findings. ***First***, direct task-specific RL fails to achieve the anticipated improvements, particularly in visual generation, and even impairs other abilities. ***Second***, compared with alternative strategies, unified RL showcases comprehensive advantages across tasks. These results demonstrate the synergistic co-evolution of dual capabilities under a shared policy optimization paradigm.

In light of our preliminary findings, we propose **CoRL**, a co-reinforcement learning framework designed to synergistically improve the understanding and generation capabilities of ULMs. Specifically, CoRL follows a *Foundation-then-Specialization* paradigm and is implemented through a two-stage RL procedure: a unified RL stage for joint optimization of dual capabilities and a refined RL stage for task-specific enhancement. In the first stage, the policy ULM is optimized through a unified GRPO algorithm with diverse rewards on a carefully curated dataset spanning both understanding and generation tasks. To effectively guide policy optimization in visual generation, we introduce a *bidirectional cycle consistency reward* and a *text-image matching reward*, which together promote semantic consistency and faithfulness between synthesized images and their corresponding prompts. The designed rewards complement typical multimodal understanding rewards (*i.e.*, accuracy and format) within a unified group, enabling cross-task joint optimization. In the subsequent stage, we independently reinforce the policy's understanding and generation capabilities using respective rewards and tailored datasets for task-specific refinement.

Applying the two-stage CoRL procedure to the baseline ULM Janus-Pro [8] yields **ULM-R1**, a unified model with reinforced capabilities in both understanding and generation. To comprehensively assess its performance, we conduct extensive comparisons against state-of-the-art unified MLLMs and dedicated models across both three visual generation and nine multimodal understanding benchmarks. Notably, ULM-R1 achieves substantial gains over its baseline on complex mathematical and logical reasoning tasks, such as WeMath (+15.2) and LogicVista (+10.6). These results underscore the effectiveness of CoRL, providing compelling empirical evidence for the efficacy of RL in jointly advancing visual understanding and generation tasks.

We summarize our main contributions as follows:

- We establish that RL with verifiable rewards and GRPO constitutes a data-efficient paradigm for cross-task co-optimization and capability enhancement.

- We introduce a co-reinforcement learning framework, CoRL, to synergistically enhance the dual capabilities of ULMs using a unified-then-refined RL paradigm.

- We demonstrate the effectiveness of CoRL and the advantage of ULM-R1 through extensive qualitative and quantitative experiments across diverse benchmarks.

## 2 Related Work

**Unified Multimodal Understanding and Generation.** Recent advancements [8, 39, 67, 71, 76, 82, 86–88, 90, 91, 107] have witnessed increasing attention to jointly model multimodal understanding and visual generation within a unified model. Pioneering attempts [12, 17] predominantly rely on continuous diffusion models, integrating external diffusion decoders for image synthesis. Inspired by autoregressive next-token prediction, a growing line of research [8, 31, 39, 47, 57, 71, 82, 85–88, 111] encode visual inputs into discrete tokens and generate images in a *fully autoregressive* (F-AR) manner. Specifically, this approach employs a vector quantized (VQ) tokenizer [14, 93] to convert images into discrete tokens, analogous to text tokenization. To mitigate information loss in VQ discretization, another stream of work [4, 16, 23, 29, 48, 65, 74, 76, 90, 92] explores *autoregressive and diffusion* (AR-Diff) hybrid modeling approaches. Architecturally, these models typically comprise a vision autoencoder, a text tokenizer, and an LLM. Given the unified advantage of the F-AR model in generation manner, this work builds upon it to develop our co-reinforcement learning framework.

**RL-based Post-Training for MLLMs.** Post-training [108] aims to further enhance the performance of pretrained models for customized applications and user needs. Recently, RL [78, 84] has emerged as a powerful post-training technique, enabling models to learn from feedback and align with human values. RL in MLLMs can be broadly categorized into two paradigms: (1) RL from human/AI feedback (RLHF) [34, 54, 61, 68, 77, 81–83, 92, 95, 96, 99, 106, 109] and (2) RL with verifiable reward mechanisms [35, 40, 41, 69, 79, 100]. RLHF involves learning reward models from preference data before RL optimization, whereas the latter directly optimizes models using task-specific reward functions, bypassing explicit preference modeling. For example, DPO [59] is a notable implementation of RLHF and has been adopted by Emu3 [82] and HermesFlow [92] to narrow the performance gap between understanding and generation. In contrast, GRPO [64] exemplifies the second paradigm, simplifying reward formulation via group-wise relative advantage estimation. Our work also falls into this paradigm but diverges from prior work such as SimpleAR [79], which utilizes GRPO with external CLIP reward for autoregressive visual generation, and R1-like MLLMs [24, 41, 69, 100] that focus on incentivizing reasoning capabilities. First, our work demonstrates the significant potential of RL in co-optimizing understanding and generation, thereby broadening its applicability beyond reasoning. Moreover, we identify *semantic consistency rewards* and *a co-evolutionary reinforcement strategy* as crucial components in enhancing ULMs.

## 3 Methodology

### 3.1 Preliminary

Group relative policy optimization (GRPO) [64] is a value-free policy optimization algorithm with improved training stability and sample efficiency. Building upon PPO [62], GRPO introduces a group-wise relative advantage approach to bound policy updates while maintaining optimization flexibility. Let $\pi_{\boldsymbol{\theta}}$ denote a policy parameterized by $\boldsymbol{\theta}$. Formally, given an input content $c$, the algorithm first samples a group of $G$ outputs $\{o_1, o_2, \ldots, o_G\}$ from the current policy $\pi_{\boldsymbol{\theta}_{\text{old}}}$. Each output is then evaluated using predefined, verifiable reward functions, yielding the reward set $\{r_1, r_2, \ldots, r_G\}$. These rewards are subsequently normalized to compute group-relative advantages as follows:

$$A_i = \frac{r_i - \text{mean}(\{r_1, r_2, \ldots, r_G\})}{\text{std}(\{r_1, r_2, \ldots, r_G\})}. \tag{1}$$

After obtaining the advantage set $\{A_1, A_2, \ldots, A_G\}$ via group relative advantage estimation, the policy $\pi_{\boldsymbol{\theta}}$ is optimized by maximizing the following objective:

$$\mathcal{L}(\boldsymbol{\theta}) = \mathbb{E}_{\{o_i\}_{i=1}^{G} \sim \pi_{\boldsymbol{\theta}_{\text{old}}}} \frac{1}{G} \sum_{i=1}^{G} \left[ \frac{\pi_{\boldsymbol{\theta}}(o_i)}{\pi_{\boldsymbol{\theta}_{\text{old}}}(o_i)} A_i - \beta \, \mathbb{D}_{\text{KL}} \left( \pi_{\boldsymbol{\theta}} \, \| \, \pi_{\text{ref}} \right) \right], \tag{2}$$

where $\mathbb{D}_{\text{KL}}$ denotes the KL-divergence used to constrain the deviation between $\pi_{\boldsymbol{\theta}}$ and its reference policy $\pi_{\text{ref}}$, and $\beta$ is a regularization coefficient.

### 3.2 Pilot Exploration

Given the exceptional performance and data efficiency of DeepSeek-R1-Zero [19], we explore the potential of ULMs to enhance understanding and generation capabilities without dependence on task-specific supervised fine-tuning. To accomplish this, we curate a dataset[2] comprising 16K samples sourced from the COCO 2017 training split [38]. Each sample includes a real image, an associated caption as a textual prompt for visual generation, and a corresponding QA pair for the multimodal understanding task. We adopt CLIP Score [58] as the verifiable reward for image generation, along with a combination of formatting correctness and answer

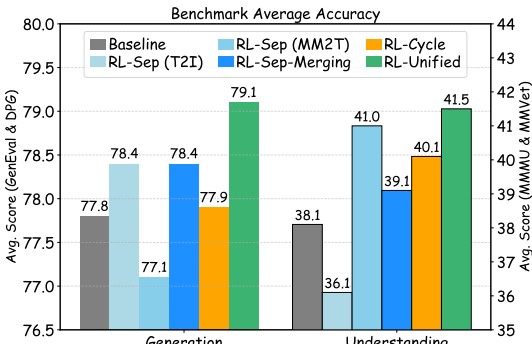

Figure 1: **Results of different RL paradigms.** Janus-Pro-1B [8] serves as the baseline.

---

[2]https://huggingface.co/datasets/mm-vl/x2x_rft_16k

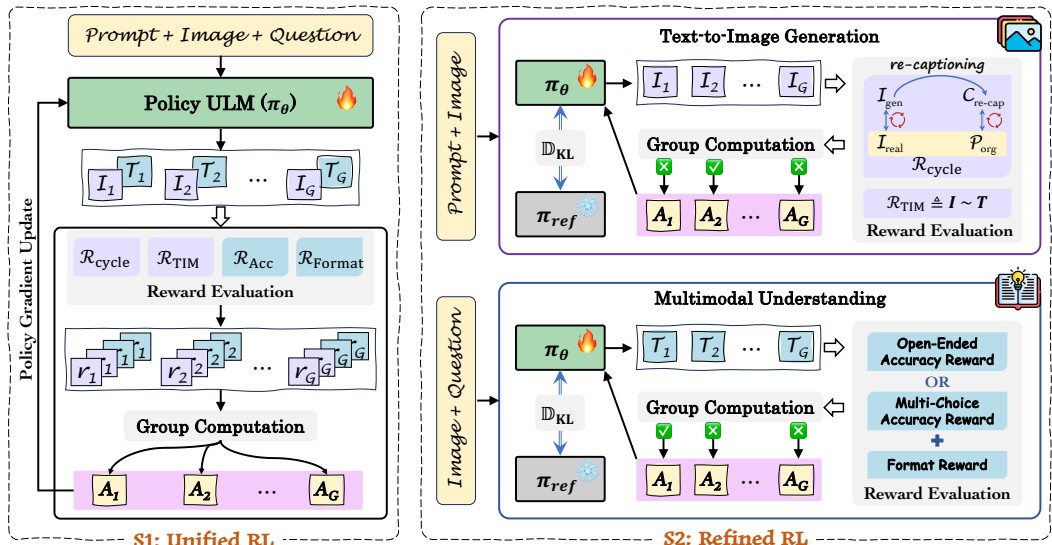

Figure 2: **Overview of CoRL**, a co-reinforcement learning framework to jointly improve the dual capabilities of ULMs. CoRL adopts a two-stage RL procedure, comprising a unified RL stage for joint optimization and a refined RL stage for task-specific enhancement.

accuracy as the reward for text generation. We investigate four distinct RL paradigms: (*i*) *separate RL*, where understanding and generation tasks are independently optimized with their respective reward mechanisms; (*ii*) *separate RL followed by weight merging*, where each task is separately optimized, and the resulting weights are subsequently merged using a Gaussian distribution-based merging strategy [66] to incorporate both abilities; (*iii*) *cycle RL*, which employs a scheduled alternation between the two tasks throughout the training process; and (*iv*) *unified RL*, in which both tasks are jointly optimized within a unified paradigm to promote the co-evolution of dual capabilities.

As presented in Figure 1, we observe that (1) direct task-specific RL fails to achieve the expected improvements for ULMs, particularly in the visual generation task, and may even impair performance on the other task; and (2) unified RL demonstrates substantial advantages over alternative paradigms. These findings indicate that the dual capabilities that co-evolve within a shared training framework contribute to enhanced cross-task synergy and knowledge transfer.

## 3.3 Co-Reinforcement Learning

### 3.3.1 Verifiable Reward for Multimodal modeling

In this section, we develop a suite of verifiable rewards for multimodal modeling, which provide clear and objective feedback to steer ULMs toward generating high-quality image and text outputs.

**Bidirectional Cycle Consistency Reward in Text-to-Image Generation.** To encourage ULMs to generate images that faithfully depict the concepts and entities described in the input prompt, we introduce a bidirectional cycle consistency reward $\mathcal{R}_{\text{cycle}}$, which measures the consistency between predictions and ground truth in both visual and textual spaces. For visual consistency, we adopt LPIPS [104] to assess the patch-level perceptual similarity between the real image $\mathcal{I}_{\text{real}}$ and the synthesized image $\mathcal{I}_{\text{gen}}$. Textual consistency is implemented in a re-captioning manner. Specifically, we first employ BLIP [32] to generate a caption $\mathcal{C}_{\text{re-cap}}$ for each synthesized image, and then compute the SPICE [1] score between $\mathcal{C}_{\text{re-cap}}$ and its original prompt $\mathcal{P}_{\text{org}}$ to measure semantic fidelity. The combined bidirectional cycle reward is defined as:

$$\mathcal{R}_{\text{cycle}} = 1 - \text{LPIPS}(\mathcal{I}_{\text{real}}, \mathcal{I}_{\text{gen}}) + \text{SPICE}(\mathcal{P}_{\text{org}}, \mathcal{C}_{\text{re-cap}}). \quad (3)$$

This bidirectional reward forms a closed feedback loop that promotes mutual consistency between texts and images, effectively penalizing hallucinated content and reinforcing prompt-aligned visual generation by simultaneously optimizing for both visual and textual consistency. Furthermore, $\mathcal{R}_{\text{cycle}}$ is normalized to the range [0, 1] before being combined to ensure that all rewards operate on comparable scales and to prevent any single component from dominating due to scale differences.

**Text-Image Matching Reward.** While CLIP Score [58] provides a holistic measure of text-image alignment, as shown in Sec. 3.2, it underperforms due to its limited capacity for assessing fine-grained semantics. To address this limitation, we instead propose a text-image matching reward $\mathcal{R}_{\text{TIM}}$, which leverages the ULM itself to evaluate cross-modal alignment at the token level. Given a textual representation $\boldsymbol{T} = \{\boldsymbol{t}_1, \boldsymbol{t}_2, \ldots, \boldsymbol{t}_{L_t}\} \in \mathbb{R}^{L_t \times d}$ of the prompt and the corresponding visual representation $\boldsymbol{I} = \{\boldsymbol{i}_1, \boldsymbol{i}_2, \ldots, \boldsymbol{i}_{L_i}\} \in \mathbb{R}^{L_i \times d}$ of a generated image, the reward is computed as:

$$\mathcal{R}_{\text{TIM}} = \frac{1}{2} \left( \frac{1}{L_i} \sum_{j=1}^{L_i} \max_{k \in [1, L_t]} \cos(\boldsymbol{i}_j, \boldsymbol{t}_k) + \frac{1}{L_t} \sum_{k=1}^{L_t} \max_{j \in [1, L_i]} \cos(\boldsymbol{t}_k, \boldsymbol{i}_j) \right), \tag{4}$$

where $L_t$ and $L_i$ are the sequence lengths of the textual and visual tokens, $d$ is the embedding dimension, and $\mathcal{R}_{\text{TIM}}$ is also be normalized to the range [0, 1]. This reward captures the fine-grained correspondence between textual concepts and visual elements through maximum cosine similarity, ensuring mutual alignment between visual tokens and their most relevant textual counterparts.

**Accuracy Reward in Multimodal Question Answering.** Accuracy rewards leverage task-specific metrics to directly evaluate the correctness of ULM predictions. We consider two accuracy rewards tailored to different question types: $\mathcal{R}_{\text{MCQ-Acc}}$ for multi-choice questions and $\mathcal{R}_{\text{OE-Acc}}$ for open-ended questions. These rewards follow a binary evaluation mechanism, assigning a value of 1 when the predicted answer (*i.e.*, the final answer parsed from within `<answer>` and `</answer>` tags) matches the ground truth and 0 otherwise.

**Format Reward in Text Generation.** To encourage ULMs to generate structured and interpretable textual responses, we adopt the format reward [19], which requires the model to enclose its thinking process inside `<think>` $\cdots$ `</think>`, and provide its final answer within `<answer>` and `</answer>` tags. The format reward $\mathcal{R}_{\text{Format}}$ returns 1 for strict compliance and 0 otherwise.

### 3.3.2 Unified Reinforcement Learning for Synergistic Multimodal Modeling

As illustrated in Figure 2, the policy ULM first undergoes unified reinforcement learning with diverse rewards across understanding and generation tasks. This unified process aims to jointly enhance its dual capabilities and establish a solid foundation for subsequent task-specific refinement.

**Reward Function and Training Objective.** To ensure diversity and complementarity in reward signals for unified multimodal modeling, we formulate a joint reward function as

$$\mathcal{R}_{\text{Uni-S1}} = \mathcal{R}_{\text{cycle}} + \mathcal{R}_{\text{TIM}} + \lambda \cdot (\mathcal{R}_{\text{Acc}} + \mathcal{R}_{\text{Format}}), \tag{5}$$

where $\lambda$ is a coefficient that balances the two types of rewards. During training, given an input prompt and an image-question pair, the policy model $\pi_{\boldsymbol{\theta}_{\text{old}}}$ first generates $G$ candidate responses, $o = \{(\mathcal{I}_1, \mathcal{T}_1), (\mathcal{I}_2, \mathcal{T}_2), \ldots, (\mathcal{I}_G, \mathcal{T}_G)\}$, each comprising a synthesized image $\mathcal{I}$ and a CoT-format solution $\mathcal{T}$. Concurrently, the joint reward function $\mathcal{R}_{\text{Uni-S1}}$ evaluates each candidate pair, yielding the reward set $r = \{r_1, r_2, \ldots, r_G\}$. These rewards are subsequently normalized according to Eq. (1) to compute the corresponding group-relative advantages $A = \{A_1, A_2, \ldots, A_G\}$. The new policy model $\pi_{\boldsymbol{\theta}}$ is then updated by maximizing the following GRPO-based objective:

$$\mathcal{L}_{\text{S1}} = \mathbb{E}_{\{o_i\}_{i=1}^G \sim \pi_{\boldsymbol{\theta}_{\text{old}}}} \frac{1}{G} \sum_{i=1}^{G} \frac{\pi_{\boldsymbol{\theta}}(o_i)}{\pi_{\boldsymbol{\theta}_{\text{old}}}(o_i)} A_i \, , \text{ where } o_i = (\mathcal{I}_i, \mathcal{T}_i). \tag{6}$$

Notably, based on empirical findings from recent work [94], we omit the KL-divergence constraint during this stage to improve both optimization efficiency and generalization capability.

**Training Data.** To support unified RL for synergistic multimodal modeling, we curate a comprehensive dataset comprising 22K samples[3], which follows the data structure established in Sec. 3.2. Each sample includes *a real image*, *a prompt* for visual generation, and *a CoT-format QA pair* for multimodal understanding. This balanced data composition facilitates joint optimization of dual capabilities within a unified framework, while preserving the granularity of task-specific supervision.

---

[3]https://huggingface.co/datasets/mm-vl/x2x_rft_22k

Table 1: **Results on text-to-image generation benchmarks.** ♣ and ♣ denote models trained using DPO and GRPO strategies. The best performance in each category is highlighted in **bold**.

| Model | Scale | Res. | Type | GenEval ↑ | | | | | WISE ↑ | DPG ↑ |
|---|---|---|---|---|---|---|---|---|---|---|
| | | | | Two Obj. | Counting | Position | Color Attri. | Overall | Overall | Overall |
| ▼ *Generation Only* | | | | | | | | | | |
| PixArt-$\alpha$ [5] | 0.6B | $512^2$ | Diff | 0.50 | 0.44 | 0.08 | 0.07 | 0.48 | **0.47** | 71.11 |
| SDv1.5 [60] | 0.9B | $512^2$ | Diff | 0.38 | 0.35 | 0.04 | 0.06 | 0.43 | 0.32 | 63.18 |
| SDv2.1 [60] | 0.9B | $512^2$ | Diff | 0.51 | 0.44 | 0.07 | 0.17 | 0.50 | 0.32 | 68.09 |
| SD3-Medium [15] | 2B | $512^2$ | Diff | **0.94** | **0.72** | **0.33** | **0.60** | **0.74** | 0.42 | **84.08** |
| SDXL [55] | 2.6B | $1024^2$ | Diff | 0.74 | 0.39 | 0.15 | 0.23 | 0.55 | 0.43 | 74.65 |
| DALL·E 3 [3] | - | $1024^2$ | Diff | 0.87 | 0.47 | 0.43 | 0.45 | 0.67 | - | 83.50 |
| LlamaGen [67] | 0.8B | $256^2$ | F-AR | 0.34 | 0.21 | 0.07 | 0.04 | 0.32 | - | 65.16 |
| SimpleAR [79] ♣ | 1.5B | $1024^2$ | F-AR | 0.90 | - | 0.28 | 0.45 | 0.63 | - | 81.97 |
| ▼ *Unified Understanding and Generation* | | | | | | | | | | |
| TokenFlow [57] | 8B | $256^2$ | F-AR | 0.60 | 0.41 | 0.16 | 0.24 | 0.55 | - | 73.38 |
| Emu3 [82] | 8B | $512^2$ | F-AR | - | - | - | - | 0.66 | 0.39 | 80.60 |
| Emu3-DPO [82] ♣ | 8B | $512^2$ | F-AR | - | - | - | - | 0.64 | - | 81.60 |
| LWM [39] | 7B | $512^2$ | F-AR | 0.41 | 0.46 | 0.09 | 0.15 | 0.47 | - | - |
| Orthus [29] | 7B | $512^2$ | AR-Diff | - | - | - | - | 0.58 | 0.27 | - |
| Janus-Pro [8] | 7B | $384^2$ | F-AR | **0.89** | 0.59 | **0.79** | **0.88** | **0.80** | 0.35 | **84.19** |
| ILLUME+ [23] | 3B | $384^2$ | AR-Diff | 0.88 | 0.62 | 0.42 | 0.53 | 0.72 | - | - |
| D-DiT [37] | 2B | $512^2$ | Diff | 0.80 | 0.54 | 0.32 | 0.50 | 0.65 | - | - |
| Harmon [87] | 1.5B | $512^2$ | F-AR | 0.86 | 0.66 | 0.74 | 0.48 | 0.76 | **0.41** | - |
| show-o [90] | 1.3B | $512^2$ | AR-Diff | 0.80 | 0.66 | 0.31 | 0.50 | 0.68 | 0.35 | 67.48 |
| HermesFlow [92] ♣ | 1.3B | $512^2$ | AR-Diff | 0.84 | 0.66 | 0.32 | 0.52 | 0.69 | - | 70.22 |
| Janus [85] | 1.3B | $384^2$ | F-AR | 0.68 | 0.30 | 0.46 | 0.42 | 0.61 | 0.23 | 79.68 |
| Janus-Pro [8] | 1.5B | $384^2$ | F-AR | 0.82 | 0.51 | 0.65 | 0.56 | 0.73 | 0.26 | 82.63 |
| **ULM-R1** ♣ | 1.5B | $384^2$ | F-AR | 0.85 | **0.71** | 0.68 | 0.80 | 0.77 | 0.33 | 83.92 |

### 3.3.3 Refined Reinforcement Learning for Task-specific Enhancement

After completing unified RL, as shown in Figure 2, we apply a targeted learning strategy to further enhance the task-specific performance of the policy model. This second-stage optimization leverages task-specific rewards and tailored datasets for individual tasks.

**Reward Function and Training Objective.** For text-to-image generation, the reward is defined as $\mathcal{R}_{\text{T2I-S2}} = \mathcal{R}_{\text{cycle}} + \mathcal{R}_{\text{TIM}}$. For multimodal understanding, we define two distinct reward formulations: (1) $\mathcal{R}_{\text{MCQ-S2}} = \mathcal{R}_{\text{MCQ-Acc}} + \mathcal{R}_{\text{Format}}$ for multiple-choice questions, and (2) $\mathcal{R}_{\text{OE-S2}} = \mathcal{R}_{\text{OE-Acc}} + \mathcal{R}_{\text{Format}}$ for open-ended questions. The training objective in this stage adheres to the standard GRPO formulation in Eq. (2), with the appropriate task-specific reward ($\mathcal{R}_{\text{T2I-S2}}$, $\mathcal{R}_{\text{MCQ-S2}}$, or $\mathcal{R}_{\text{OE-S2}}$) replacing $A_i$ depending on the task. To ensure stable optimization, we reintroduce the KL-divergence constraint at this stage to limit policy deviation from the reference distribution.

**Training Data.** For text-to-image generation, we continue training on the curated dataset introduced in Sec. 3.2. For multimodal understanding, we utilize two specialized datasets: mcot_r1_mcq[4] for multiple-choice questions and mcot_r1_vqa[5] for open-ended questions. These task-specific datasets enable the model to develop more refined and robust capabilities within each task domain.

## 4 Experiment

### 4.1 Experimental Setups

**Evaluation Benchmarks.** We evaluate visual generation capabilities on the GenEval [18], WISE [50], and DPG-Bench [22] benchmarks. GenEval employs an object-centric evaluation protocol to assess compositional and attribute-level alignment, while DPG-Bench adopts a VQA-based setting to evaluate dense prompt-following and semantic fidelity. WISE provides a holistic evaluation of models' world knowledge, considering consistency, realism, and aesthetics. We also evaluate multimodal understanding capabilities across diverse benchmarks. Specifically, MMStar [6], MMMU [98], and

---

[4] https://huggingface.co/datasets/mm-vl/mcot_r1_mcq_66k
[5] https://huggingface.co/datasets/mm-vl/mcot_r1_vqa_66k

Table 2: **Results on multimodal understanding benchmarks.** The best performance within each category is highlighted in **bold**. † denotes results obtained from our evaluation.

| Model | LLM | Multi-Choice (MC) ↑ | | | Open-Ended (OE) ↑ | | | MC&OE Mixed ↑ | | |
|---|---|---|---|---|---|---|---|---|---|---|
| | | MMMU | MMStar | Math$^{We}$ | MMVet | POPE | Logic$^{VT}$ | Math$^{VT}$ | Math$^{VS}$ | Math$^{Vis}$ |
| ▼ *Understanding Only* | | | | | | | | | | |
| SmolVLM [49] | SmolLM2-1.7B | 38.8 | 41.7 | 9.1 | 33.8 | 85.5 | 28.0 | 43.6 | 12.6 | 12.8 |
| SAIL-VL [11] | Qwen2.5-1.5B | 44.1 | 56.5 | 14.6 | 44.2 | 88.1 | 30.4 | 62.8 | 17.4 | 17.3 |
| Ovis2 [45] | Qwen2.5-1.5B | 45.6 | 56.7 | 9.9 | 58.3 | 87.8 | 34.7 | **64.1** | 29.4 | 17.7 |
| InternVL3 [110] | Qwen2.5-1.5B | 48.7 | **61.1** | **22.9** | **67.0** | 90.1 | 34.7 | 57.6 | 24.5 | 20.2 |
| Qwen2.5-VL [2] | Qwen2.5-3B | **51.2** | 56.3 | **22.9** | 60.0 | 85.9 | **40.3** | 61.2 | 31.2 | 21.9 |
| LMM-R1 [53] | Qwen2.5-3B | - | 58.0 | - | - | - | - | 63.2 | **41.6** | **26.4** |
| ▼ *Unified Understanding and Generation* | | | | | | | | | | |
| ILLUME+ [23] | Qwen2.5-3B | **44.3** | - | - | 40.3 | 87.6 | - | - | - | - |
| Harmon [87] | Qwen2.5-1.5B | 38.9 | - | - | - | 87.6 | - | - | - | - |
| VILA-U [88] | LLaMA-2-7B | - | - | - | 33.5 | 85.8 | - | - | - | - |
| Orthus [29] | Chameleon-7B | 28.2 | - | - | - | 79.6 | - | - | - | - |
| UniToken [27] | Chameleon-7B | 32.8 | 46.1 | - | - | - | - | 38.5 | - | |
| SGen-VL [31] | InternLM2-1.8B | 34.2 | - | - | 34.5 | 85.3 | - | **42.7** | - | |
| Show-o [90] | Phi-1.3B | 26.7 | - | - | - | 80.0 | - | - | - | - |
| HermesFlow [92] | Phi-1.3B | 28.3 | - | - | - | 81.4 | - | - | - | - |
| Janus-Pro [8] | DeepSeek-LLM-7B | 41.0 | 46.5 | 9.7 | **50.0** | 87.4 | 28.0 | 42.5 | 15.9 | 14.7 |
| Janus [85] | DeepSeek-LLM-1.3B | 30.5 | 37.6 | 3.4$^\dagger$ | 34.3 | 87.0 | 23.9$^\dagger$ | 33.7 | 14.9$^\dagger$ | 13.4$^\dagger$ |
| Janus-Pro [8] | DeepSeek-LLM-1.5B | 36.3 | 43.1$^\dagger$ | 5.9$^\dagger$ | 39.8 | 86.2 | 23.9$^\dagger$ | 37.3$^\dagger$ | 13.5$^\dagger$ | 13.4$^\dagger$ |
| **ULM-R1** | DeepSeek-LLM-1.5B | 42.3 | **47.6** | 21.1 | 43.9 | **88.9** | 34.5 | 42.5 | **25.4** | **22.0** |

WeMath (Math$^{We}$) [56] are used for multi-choice evaluation, while MMVet [97], POPE [36], and LogicVista (Logic$^{VT}$) [89] are used for open-ended evaluation. In addition, we employ MathVista (Math$^{VT}$) [43], MathVerse-Vision (Math$^{VS}$) [102], and MathVision (Math$^{Vis}$) [80] to assess complex mathematical reasoning capabilities, covering both multi-choice and open-ended QA formats. On these benchmarks, we compute accuracy using the toolkit VLMEvalKit [13].

**Implementation Details.** We develop ULM-R1 using Janus-Pro-1B [8] as the baseline ULM for unified multimodal understanding and generation. To ensure reproducibility and scalability, our RL training is built upon the trl [75] framework. In the unified RL stage, we employ the AdamW optimizer with an initial learning rate of 4e-6 and a batch size of 16. We sample 8 responses for both understanding and generation tasks, and set the reward balancing factor in Eq. (5) to 0.8. In the refined RL stage, we sample 16 responses for both multimodal understanding and text-to-image generation tasks. Additionally, we reduce the learning rate to 1e-6 to facilitate fine-grained optimization. All training is conducted on 8 NVIDIA H20 (96G) GPUs. During inference, greedy decoding is used for text generation in multimodal understanding tasks. For text-to-image generation, we employ classifier-free guidance (CFG) [20] with a guidance weight set to 5. More details on the training data and settings are provided in App. A.

### 4.2 Quantitative Results

**Text-to-Image Generation.** Table 1 presents a comprehensive comparison between ULM-R1 and state-of-the-art models across three visual generation benchmarks. Among unified models, our model ranks second on both GenEval and WISE benchmarks. Notably, it achieves balanced performance across diverse task categories within GenEval, with the best score of 0.71 in object counting. When compared with specialized generation-only models, ULM-R1 surpasses the top performer SD3-Medium [15] by a slight margin (0.77 *vs.* 0.74 on GenEval). Moreover, ULM-R1 shows consistent improvements over its base model across all benchmarks. These results collectively demonstrate the effectiveness and advantage of our CoRL in enhancing visual generation quality.

**Multimodal Understanding.** Results are shown in Table 2. For mixed QA format evaluation, we continue to apply the Gaussian-distribution-based merging strategy [66] to combine the two task-specific policy models and obtain a final model capable of following both types of instructions. Overall, ULM-R1 markedly outperforms existing unified models across most benchmarks, and substantially narrows the performance gap with leading understanding-only MLLMs of comparable model scale. More specifically, our model achieves state-of-the-art performance among unified models

Table 3: **Comparison between different RL paradigms for ULMs.** The cold SFT data is consist of `x2x_rft_22k`, `mcot_r1_mcq` (22K), and `mcot_r1_vqa` (22K). #7: CoRL.

| # | Ablated Setting | Stage | GenEval | DPG | MMMU | Math$^{We}$ | MMVet | Logic$^{VT}$ |
|---|---|---|---|---|---|---|---|---|
| 0 | Baseline | - | 73.0 | 82.6 | 36.3 | 5.9 | 39.8 | 23.9 |
| 1 | + Cold-SFT | S1 | 72.8 (-0.3) | 82.5 (-0.1) | 41.0 (+4.7) | 18.0 (+12.1) | 42.0 (+2.2) | 27.9 (+4.0) |
| 2 | + Unified-RL | S1 | 75.9 (+2.9) | 83.3 (+0.7) | 40.3 (+4.0) | 14.0 (+8.1) | 42.5 (+2.7) | 30.2 (+6.3) |
| 3 | + Refined-RL (T2I) | S2 | 75.1 (+2.1) | 83.0 (+0.4) | / | / | / | / |
| 4 | + Refined-RL (MM2T-MC) | S2 | / | / | 39.6 (+3.3) | 15.8 (+9.9) | / | / |
| 5 | + Refined-RL (MM2T-OE) | S2 | / | / | / | / | 42.2 (+2.4) | 29.5 (+5.6) |
| 6 | + Refined-RL w/ Cold-SFT | S1&S2 | 74.5 (+1.5) | 82.8 (+0.2) | 41.8 (+5.5) | 22.5 (+16.6) | 43.7 (+3.9) | 35.9 (+12.0) |
| 7 | + Refined-RL w/ Unified-RL | S1&S2 | 77.3 (+4.3) | 83.9 (+1.3) | 42.3 (+6.0) | 21.1 (+15.2) | 43.9 (+4.1) | 34.5 (+10.6) |

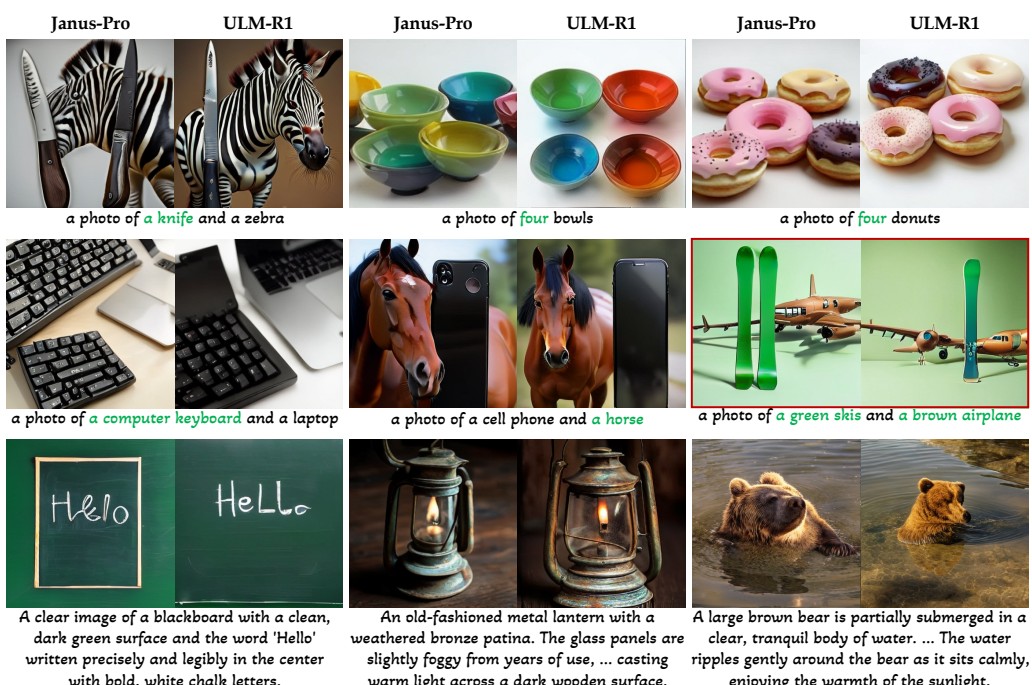

| Janus-Pro | ULM-R1 | Janus-Pro | ULM-R1 | Janus-Pro | ULM-R1 |

a photo of *a knife* and a zebra — a photo of *four* bowls — a photo of *four* donuts

a photo of *a computer keyboard* and a laptop — a photo of a cell phone and *a horse* — a photo of *a green skis* and *a brown airplane*

A clear image of a blackboard with a clean, dark green surface and the word 'Hello' written precisely and legibly in the center with bold, white chalk letters.

An old-fashioned metal lantern with a weathered bronze patina. The glass panels are slightly foggy from years of use, ... casting warm light across a dark wooden surface.

A large brown bear is partially submerged in a clear, tranquil body of water. ... The water ripples gently around the bear as it sits calmly, enjoying the warmth of the sunlight.

Figure 3: **Qualitative comparison of text-to-image generation** between Janus-Pro and ULM-R1. The red box marks an exemplary failure case.

on MMStar (47.6), WeMath (21.1), LogicVista (34.5), and on several mixed-format math benchmarks, including MathVerse (25.4) and MathVision (22.0). Particularly, ULM-R1 demonstrates considerable improvements over its base model in mathematical and logical reasoning tasks, achieving gains of **15.2** on WeMath and **10.6** on LogicVista. These results not only demonstrate the effectiveness of CoRL in enhancing ULMs' understanding capabilities, but also establish that reinforcement learning provides a data-efficient pathway for achieving both robust generalization and sophisticated reasoning capabilities, without the need for large-scale supervised data.

## 4.3 Qualitative Results

In this section, we first present a qualitative comparison between ULM-R1 and Janus-Pro for visual generation, as illustrated in Figure 3. The results clearly show that ULM-R1 achieves superior text-to-image alignment and object grounding across diverse prompts, with especially notable improvements in spatial arrangement of objects and compositional consistency. Next, as shown in Figure 4, we visualize several representative examples of multimodal understanding. Compared to Janus-Pro, ULM-R1 exhibits significantly enhanced understanding capabilities, particularly in mathematical reasoning. These comprehensive qualitative results demonstrate the effectiveness of CoRL in simultaneously

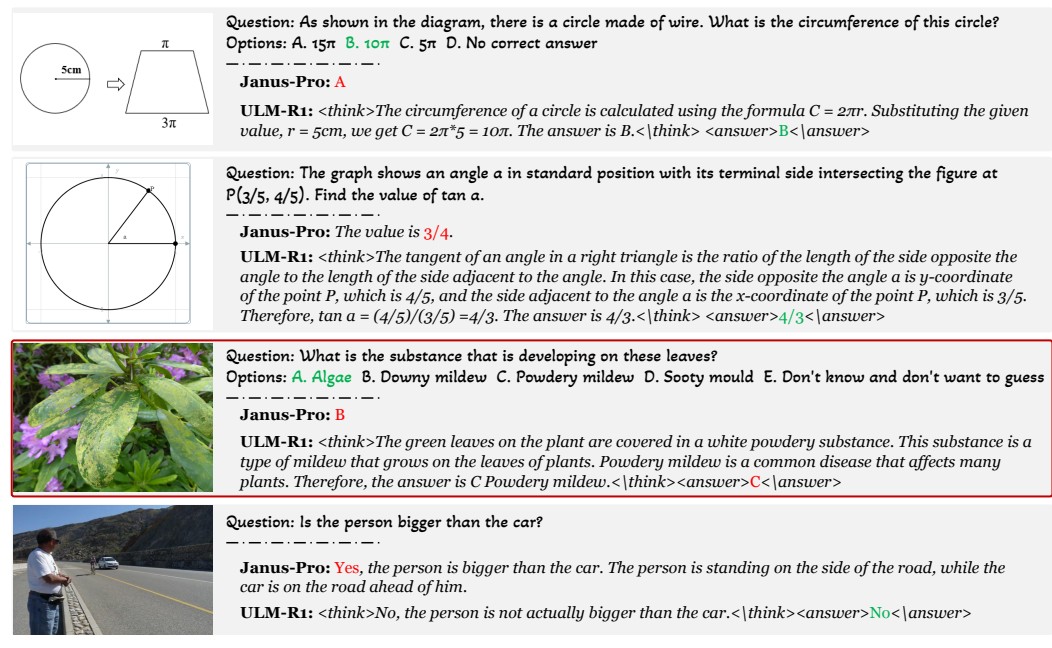

Figure 4: **Qualitative comparison of multimodal understanding** between Janus-Pro and ULM-R1. The red box marks an exemplary failure case.

improving visual generation and multimodal understanding in ULMs. Furthermore, Figures 3 and 4 respectively showcase exemplary failure cases of ULM-R1 in visual generation and understanding tasks, providing an intuitive grasp of its limitations. For instance, in the understanding example, it misinterprets commonsense and professional knowledge, leading to an incorrect answer.

### 4.4 Ablation Study and Discussion

In this section, we primarily evaluate the effectiveness of our RL training strategy and the proposed reward functions for text-to-image generation. In addition, we discuss the impact of the hyperparameter $\lambda$ and the scalability of CoRL.

**Comparison Between Various RL Paradigms.** As presented in Table 3, we conduct a comprehensive ablation study to evaluate the effects of different RL paradigms for ULMs. The results reveal two key findings: ▶ #2 *vs.* #1: Unified-RL effectively enhances both the generation and understanding capabilities of ULMs, whereas Cold-SFT has minimal impact on visual generation. ▶ #7 *vs.* #6: Compared to the de facto paradigm, our CoRL consistently outperforms it on visual generation benchmarks while achieving comparable results on multimodal understanding benchmarks. These findings indicate that *unified RL provides a robust foundation for task-specific refinement, even without reliance on supervised data*. Additionally, CoRL consistently outperforms both its baseline and task-specific RL variants (#3-#5), achieving improvements of 2.1 points on GenEval (*vs.* generation-only RL, #3) and 5.3 points on WeMath (*vs.* understanding-only RL, #4). These results demonstrate the efficacy of CoRL as our final RL paradigm.

**Effect of Rewards in Text-to-Image Generation.** To evaluate the effectiveness of our proposed rewards for text-to-image generation, we conduct ablation experiments as detailed in Table 4. The results demonstrate that incorporating either reward individually improves performance over the baseline: $\mathcal{R}_{\text{cycle}}$ yields an increase of 2.1 in average score, while $\mathcal{R}_{\text{TIM}}$ results in an increase of 0.8. Notably, combining both rewards leads to the best overall performance, achieving an average score of 80.6. These findings suggest a modest but complementary effect between $\mathcal{R}_{\text{cycle}}$ and $\mathcal{R}_{\text{TIM}}$, enhancing their joint benefit in enhancing visual generation quality. In addition, we further compare the CLIP score ($\mathcal{R}_{\text{CLIP}}$) and $\mathcal{R}_{\text{TIM}}$ under our final RL training paradigm. As shown in the table, $\mathcal{R}_{\text{TIM}}$ achieves better overall performance, especially on the DPG benchmark with dense, long-horizon prompts for image generation, highlighting its superior ability to capture fine-grained semantic alignment compared to the CLIP score.

Table 4: Effect of visual generation rewards.

| Rewards | GenEval | DPG | Avg. ↑ |
|---|---|---|---|
| Baseline | 73.0 | 82.6 | 77.8 |
| $\mathcal{R}_{\text{CLIP}}$ | 74.2 | 82.4 | 78.3 (+0.5) |
| $\mathcal{R}_{\text{TIM}}$ | 74.1 | 83.0 | 78.6 (+0.8) |
| $\mathcal{R}_{\text{cycle}}$ | 76.2 | 83.5 | 79.9 (+2.1) |
| $\mathcal{R}_{\text{cycle}} + \mathcal{R}_{\text{CLIP}}$ | 77.0 | 83.4 | 80.2 (+2.4) |
| $\mathcal{R}_{\text{cycle}} + \mathcal{R}_{\text{TIM}}$ | 77.3 | 83.9 | 80.6 (+2.8) |

Table 5: Comparison among visual consistency measures used in $\mathcal{R}_{\text{cycle}}$.

| Measures | GenEval | DPG |
|---|---|---|
| PSNR | 76.0 | 82.4 |
| MSE | 77.1 | 83.2 |
| SSIM | 77.5 | 83.6 |
| LPIPS | 77.3 | 83.9 |

Table 6: Impact of $\lambda$.

| $\lambda$ | GenEval (Gen.) | MMMU (Und.) |
|---|---|---|
| 0.5 | 77.1 | 41.0 |
| 0.7 | 77.3 | 42.3 |
| 0.8 | 77.3 | 42.3 |
| 0.9 | 77.1 | 43.5 |
| 1.0 | 76.9 | 43.0 |

**Impact of Visual Consistency Measures in $\mathcal{R}_{\text{cycle}}$.** Table 5 provides a more detailed analysis of how different visual consistency measures (PSNR, MSE, SSIM, and LPIPS) used in $\mathcal{R}_{\text{cycle}}$ affect the quality of visual generation. PSNR and MSE are pixel-level metrics that quantify low-level differences between the generated images, while SSIM and LPIPS assess higher-level perceptual and structural similarities. As shown in the table, SSIM and LPIPS perform better than the other two metrics, with LPIPS achieving the best performance (83.9) on the DPG benchmark. This can be attributed to the fact that LPIPS measures image similarity in a feature space, making it more robust to minor, semantically irrelevant variations and thus better suited to reward high-level consistency.

**Impact of hyperparameter $\lambda$.** The factor $\lambda$ in Eq. (5) balances the reward scales between the two tasks during unified RL. As shown in Table 6, we conduct experiments using different values of $\lambda$ to assess its impact on both generation and understanding performance. The results show that moderate values of $\lambda$ ($\sim 0.8$) achieve a balanced trade-off between generation and understanding. Larger values slightly degrade generation performance, indicating that overemphasizing understanding rewards may hinder cross-task optimization.

**Scalability of CoRL.** To validate the effectiveness of CoRL on other ULMs, as illustrated in Table 7, we conduct additional experiments using Janus-1.3B [85] and Janus-Pro-7B [8] as the baseline. The results show consistent improvements across both generation and understanding benchmarks, confirming the scalability of CoRL. Notably, Janus-Pro-7B with LoRA tuning achieves smaller gains on the mathematical reasoning benchmark (WeMath) than Janus-1.3B, suggesting that while CoRL scales well across model size, its enhancement of complex reasoning does not scale linearly.

Table 7: **Effectiveness of CoRL on other ULMs.** For Janus-Pro-7B, we adopt LoRA tuning [21] to enable efficient training and mitigate memory pressure during unified RL.

| Methods | GenEval | WISE | DPG | MMMU | MMStar | Math[We] | MMVet | POPE | Logic[VT] |
|---|---|---|---|---|---|---|---|---|---|
| Janus-1.3B | 0.61 | 0.23 | 79.68 | 30.5 | 37.6 | 3.4† | 34.3 | 87.0 | 23.9 |
| + CoRL (Full Fine-Tuning) | 0.64 | 0.26 | 80.92 | 34.6 | 41.9 | 16.4 | 36.9 | 88.1 | 27.0 |
| Janus-Pro-7B | 0.80 | 0.35 | 84.19 | 41.0 | 46.5 | 9.7 | 50.0 | 87.4 | 28.0 |
| + CoRL (LoRA Tuning) | 0.82 | 0.41 | 84.97 | 44.6 | 49.5 | 16.0 | 52.6 | 88.0 | 32.4 |

# 5 Limitation

Despite the substantial improvements achieved, several limitations remain that warrant further investigation. First, a notable performance gap still exists between generation and understanding tasks of ULMs. Second, our rewards for multimodal understanding are relatively simple and primary. These limitations highlight the need for more sophisticated RL designs that can further enhance understanding capabilities and narrow the performance gap. We hope our work provides valuable insights for future RL research in ULMs.

# 6 Conclusion

In this work, we investigate how to jointly enhance the understanding and generation capabilities of ULMs, and propose a co-reinforcement learning framework (CoRL). Within the proposed CoRL, the policy model follows a Foundation-then-Specialization paradigm that involves a two-stage RL procedure: a unified RL stage for joint optimization and a refined RL stage for task-specific enhancement, yielding ULM-R1. Extensive evaluations across diverse understanding and generation benchmarks demonstrate the effectiveness of CoRL and the advantage of ULM-R1.

**Acknowledgements.** This work was supported in part by NSFC (62406189, 62322113, 62376156), Shanghai Municipal Science and Technology Major Project (2021SHZDZX0102), and the Fundamental Research Funds for the Central Universities.

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

# A Appendix

## A.1 Training Data

**Training Data for Unified Reinforcement Learning.** To support synergistic multimodal modeling during unified RL, we curate a dataset (*i.e.*, x2x_rft_22k) that simultaneously involves text-to-image generation and multimodal understanding tasks. As illustrated in Figure 5, each sample includes *a real image*, *a prompt* for generation, and *a problem* for understanding. The real images are sourced from the COCO 2017 train split [38], while the problems and their corresponding solutions are adapted from A-OKVQA [63] and GPT-VQA [105]. In addition, prompts are selected from the original COCO captions based on their entity coverage with the problem solutions.

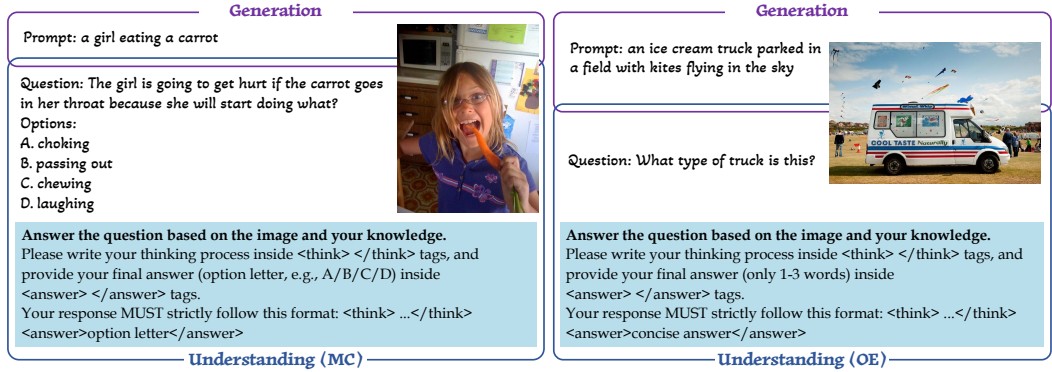

Figure 5: **Illustration of training examples used in unified reinforcement learning.**

**Training Data for Refined Reinforcement Learning.** In this stage, we collect three specialized datasets for task-specific RL. For text-to-image generation, we continue constructing a dataset (*i.e.*, x2x_rft_16k) with prompts derived from COCO captions. Moreover, we curate mcot_r1_mcq and mcot_r1_vqa for multiple-choice and open-ended multimodal understanding, respectively. These two datasets are curated on top of MCoT-Instruct [26], which encompasses a diverse range of multi-modal tasks, including mathematical reasoning, science-problem solving, and visual commonsense reasoning, across multiple source datasets. Specifically, the source datasets of mcot_r1_mcq comprise A-OKVQA [63], M$^3$CoT [7], SQA-IMG (train) [42], ArxivQA [33], TabMWP (MC) [44], and MAVIS-Instruct (MC) [103], while the source datasets of mcot_r1_vqa include GeomVerse [28], R-CoT [10], TabMWP (OE) [44], and MAVIS-Instruct (OE) [103].

## A.2 Supplementary Experimental Setups

Table 8 provides detailed hyperparameter settings for ULM-R1's RL training.

Table 8: **Training hyperparameter setting.**

| Configuration | Unified RL | Refined RL (T2I) | Refined RL (MM2T-MC) | Refined RL (MM2T-OE) |
|---|---|---|---|---|
| Number of sampled outputs ($G$) | 8 | 16 | 16 | 16 |
| Regularization coefficient of $\mathbb{D}_{KL}$ ($\beta$) | 0 | 0.02 | 0.02 | 0.02 |
| Max prompt length | 1024 | 256 | 1024 | 1024 |
| Max completion length | 512 | / | 512 | 512 |
| Batch size | 16 | 16 | 32 | 32 |
| Peak learning rate | 4e-6 | 1e-6 | 1e-6 | 1e-6 |
| Epoch | 1 | 1 | 1 | 1 |

