# OpenReview forum: "Co-Reinforcement Learning for Unified Multimodal Understanding and Generation"
_NeurIPS.cc/2025/Conference — NeurIPS 2025 spotlight_

### Official Review · Reviewer_hSgy · 2025-06-26

**Clarity:** 3
**Significance:** 3
**Originality:** 3
**Rating:** 4
**Confidence:** 4

**Summary:**

The paper introduces ​​CoRL (Co-Reinforcement Learning)​​, a novel two-stage reinforcement learning framework designed to synergistically enhance ​​multimodal understanding​​ and ​​generation​​ (e.g., text-to-image synthesis) capabilities within unified multimodal large models (ULMs). This work adopts GRPO as the reinforcement learning method and finds that jointly optimizing the two tasks yields better performance compared to optimizing each task separately. The final model achieves state-of-the-art results on multiple benchmarks.

**Questions:**

As discussed above, overall, this is a paper on a unified model + GRPO. Since the final results are indeed promising, I consider it a paper that meets the passing standard. The main issue lies in the Stage 2 task-specific RL setup, which contradicts the core observations of the paper. If I have misunderstood this aspect, I would welcome further discussion with the authors.

**Ethical Concerns:**

["NO or VERY MINOR ethics concerns only"]

**Final Justification:**

The author's response has addressed my concerns, and I will maintain my original score of 4.

**Limitations:**

yes

**Paper Formatting Concerns:**

There are no obvious formatting issues in the paper.

**Quality:**

4

**Strengths And Weaknesses:**

**Strengths**
1. This paper introduces two novel rewards — bidirectional cycle consistency reward and text-image matching reward. Ablation studies further demonstrate that both rewards play a crucial role in enhancing the model’s text-to-image generation capabilities.
2. This paper compares the performance of joint optimization versus separate optimization, aligning with the original intention of the unified generation model — achieving synergistic optimization between multimodal understanding and image generation, where the two tasks mutually enhance each other.
3. The paper is well-written, with clear illustrations and well-designed tables.

**Weaknesses**
1. The two-stage RL setup in this work seems somewhat peculiar. While the paper advocates for a unified optimization approach, and both ablation studies and pilot exploration demonstrate that unified optimization outperforms separate optimization, stage 2 nevertheless employs task-specific refined reinforcement learning. This inconsistency with the paper’s main thesis is somewhat puzzling.
2. The paper lacks certain ablation studies, particularly regarding the configuration of the R-cycle. For instance, why was LPIPS chosen as the reward metric for image similarity, rather than alternatives like PSNR, SSIM, or even MSE? Experimental validation supporting this choice would strengthen the argument.
3. The Related Work section of this paper lacks citations to some relevant papers, such as MetaMorph, LlamaFusion, OmniMamba and BLIP3-o. Including references to these papers would help make this section more comprehensive.

---

> ### Author Rebuttal · Authors · 2025-07-31
>
> Dear Reviewer hSgy,
>
> We sincerely thank you for your thorough review, thoughtful feedback, and positive remarks! We greatly appreciate your recognition of our novel reward designs, significant results, and clear writing. We respond to your concerns and questions below.
>
> ---
>
> > **W1 & Q1: Two-stage RL setup.**
>
> **R1:** Thank you for raising this excellent point. It gives us the opportunity to clarify the core motivation behind the two-stage design of CoRL, which we view as a key strength of our framework rather than a contradiction. Specifically, our two-stage CoRL framework is easily understood through  a `Foundation-then-Specialization` paradigm, which is both principled and practical for building powerful and versatile unified MLLMs.
>
> - **Stage 1 (Unified RL): The Foundation.** The main goal of this stage is not to outperform our task-specific method or existing leading MLLMs on the target benchmarks, but to build a powerful generalist foundation. By synergistically optimizing both understanding and generation tasks, this stage endows the unified MLLMs with balanced and comprehensive capabilities. As shown in our ablation and pilot experiments, this unified RL training provides an overall accuracy boost across the board.
>
> - **Stage 2 (Task-specific/Refined RL): The Specialization.** This second stage is an effective and crucial specialization step, demonstrating the flexibility and extensibility of our CoRL framework across a wide variety of diverse target tasks. The Math$^\text{We}$ benchmark (with text-rich geometric figures in images) is a perfect example, as it differs significantly from the unified RL data in terms of image types and reasoning types.
>
>     | Method                          | Math$^\text{We}$ | Hints                                                            |
>     |---------------------------------|------------------|------------------------------------------------------------------|
>     | Baseline                        | 5.9              | `Starting point`                                                 |
>     | + Unified RL                    | 14.0 (+8.1)      | `Stage 1 builds a strong foundation.`                            |
>     | + Task-specific RL              | 15.8 (+9.9)      | `Specialization without a strong foundation is limited.`         |
>     | + Unified RL & Task-specific RL | 21.1 (+15.2)     | `Specialization on a strong foundation yields the best results.` |
>
>     $\blacktriangleright$ The results clearly show that our two-stage paradigm is far superior to task-specific methods. This demonstrates that the unified RL foundation provides a much higher starting point for task-specific adaptation.
>
> In summary, we argue that this two-stage paradigm effectively resolves the tension between general capability and specialized excellence. It offers a scalable and practical pathway for extending unified MLLMs to a diverse and continually expanding range of applications. We will revise our paper to make this motivation clearer.
>
> > **W2: Ablation study on $\mathcal{R}_{\text{cycle}}$.**
>
> **R2:** Thank you for this valuable suggestion. We selected LPIPS for $\mathcal{R}_{\text{cycle}}$ primarily because it aligns better with human perceptual judgments than pixel-level metrics such as PSNR and SSIM. LPIPS measures image similarity in a feature space, making it more robust to minor, semantically irrelevant variations and better suited for rewarding high-level semantic consistency. Due to the short rebuttal period, we were only able to provide a comparison between LPIPS and MSE, but we will add a complete ablation study including PSNR and SSIM in the camera-ready version.
>
> |       | GenEval | DPG   | WISE |
> |-------|---------|-------|------|
> | MSE   | 0.77    | 83.15 | 0.30 |
> | LPIPS | 0.77    | 83.92 | 0.33 |
>
> As shown in the table, LPIPS achieves better performance on both the DPG and WISE benchmarks compared to MSE, indicating its superiority in measuring semantic consistency.
>
> > **W3: More related work.**
>
> **R3:** Thank you for pointing out these relevant works. We have updated the Related Work section to include them. Specifically, we have added MetaMorph, LMFusion, and BLIP3-o to `AR-Diff`, and OmniMamba to `F-AR`.
>
> ---
>
> We hope these clarifications address your concerns. Thank you again for your constructive feedback and affirmation of our paper.

---

> > ### Comment · Reviewer_hSgy · 2025-08-01
> > **Reply to the rebuttal**
> >
> > Thanks for your response, which has addressed my concerns, and I will maintain my original score of 4.

---

### Official Review · Reviewer_wJWf · 2025-07-02

**Clarity:** 3
**Significance:** 3
**Originality:** 3
**Rating:** 4
**Confidence:** 4

**Summary:**

This paper explores how to use reinforcement learning to improve both generation and understanding in unified MLLMs. The authors propose a method called CoRL, which uses a shared policy optimization process to train both capabilities together. CoRL includes two stages: a unified RL stage that jointly improves both generation and understanding, and a refined RL stage that focuses on specific tasks. Based on this approach, they train a model called ULM-R1, which shows clear improvements on T2I generation tasks and multimodal understanding benchmarks.

**Questions:**

Same as weaknesses, if the authors can resolve my concern from the weaknesses, especially the first one, I will raise my score.

**Ethical Concerns:**

["NO or VERY MINOR ethics concerns only"]

**Final Justification:**

I have read the rebuttals from the authors, and after discussion with authors, I decide to raise my score.

**Limitations:**

yes

**Quality:**

3

**Strengths And Weaknesses:**

**Strengths**:
1. The paper proves that jointly optimizing both generation and understanding with RL rewards designed for both yields better performance than other alternatives, which is intuitive and a step towards unified MLLMs.
2. The experiment results show great improvements compared to the base Janus-pro model, especially in understanding task.

**Weaknesses**:
1. In ablation study (Table 3), the SFT cold-start baseline only makes use of understanding data, which can barely help the generation task. The results for comparison on 1&2 and 6&7 seem expected, and they have similar performance on understanding tasks, sometimes SFT cold-start even has better results. The unfairness of the comparison makes the performance gain of unified-RL suspicious.
2. The experiments only are only conducted on 1.5 B model, the scalability of the method is not verified.

---

> ### Author Rebuttal · Authors · 2025-07-31
>
> Dear Reviewer wJWf,
>
> Thank you for your comprehensive review and for recognizing the intuitive nature and significance of our work. We are particularly grateful for your clear feedback and the opportunity to resolve your concerns, especially regarding the fairness of the SFT cold-start baseline. We respond to your concerns and questions below.
>
> ---
>
> > **W1: Concerns on SFT cold-start baseline.**
>
> **R1:** Thank you for raising this important point. We would like to clarify that the SFT cold-start baseline was indeed trained on both understanding and generation tasks to ensure a relatively fair comparison.
>
> - **Training Data.** As described in the caption of Table 3, the SFT cold-start data consists of x2x_rft_22k, mcot_r1_mcq (22K), and mcot_r1_vqa (22K). Among them, x2x_rft_22k is the same dataset used in our unified RL training and includes both generation and understanding tasks. Therefore, the SFT cold-start baseline does not only make use of understanding data. However, our initial attempts showed that using only x2x_rft_22k as the SFT cold-start data was almost ineffective. Even worse, it significantly degraded the model’s multimodal understanding capabilities. To address this, we ultimately used x2x_rft_22k, mcot_r1_mcq (22K), and mcot_r1_vqa (22K).
>
> - **Training Objective.** During the SFT cold-start stage, we follow Janus-Pro's training objective and use the cross-entropy loss to simultaneously optimize the model's generation and understanding capabilities.
>
> - **Results.** You made an astute observation. In practice, we did observe that SFT cold-start performs better than unified RL on certain understanding tasks, such as Math$^\text{We}$ and Logic$^\text{VT}$. This is likely due to task imbalance and the next-token prediction mode of autoregressive LLMs during SFT training, which is more direct and friendly for optimizing text generation than RL reward feedback.
>
> > **W2: Scalability of CoRL.**
>
> **R2:** We agree that verifying scalability is crucial. To address this, we conducted a new experiment using Janus-1.3B [1] as the baseline, given the limited time during the rebuttal period. The results show significant and consistent improvements across different types of benchmarks. We will incorporate further validation on larger-scale models in our open-source project and the camera-ready version.
>
> |            | GenEval | WISE | DPG   | MMMU | MMStar | Math$^\text{We}$ | MMVet | POPE | Logic$^\text{VT}$ |
> |------------|---------|------|-------|------|--------|------------------|-------|------|-------------------|
> | Janus-1.3B | 0.61    | 0.23 | 79.68 | 30.5 | 37.6   | 3.4†             | 34.3  | 87.0 | 23.9†             |
> | + CoRL     | 0.64    | 0.26 | 80.92 | 34.6 | 41.9   | 16.4             | 36.9  | 88.1 | 27.0              |
>
> [1] Chengyue Wu, Xiaokang Chen, et al. Janus: Decoupling visual encoding for unified multimodal understanding and generation. In CVPR, 2025.
>
> ---
>
> We hope these clarifications and additional results address your concerns. Thank you once again for your time and valuable feedback.

---

> > ### Comment · Reviewer_wJWf · 2025-08-05
> >
> > Thanks for the rebuttal from the authors. However, my concern still lies in the scalability of the proposed method to larger model, as only experiments on 1.5B or 1.3B models are conducted. To solidate the claim, I encourage the authors to conduct experiments on 7B/32B model to validate the scalability. At this stage, I will maintain my score.

---

> > > ### Author Response · Authors · 2025-08-06
> > >
> > > Thank you for your reply and suggestion. We conducted an additional experiment using Janus-Pro-7B as the baseline. Regarding its implementation, we perform RL training with LoRA [1] to support efficient training and mitigate memory pressure during unified RL training, while maintaining the same settings as in the original paper. The results show consistent improvements across both generation and understanding benchmarks, confirming the scalability of CoRL to larger-scale unified MLLMs.
> > >
> > > |                              | GenEval | WISE | DPG   | MMMU | MMStar | Math$^\text{We}$ | MMVet | POPE | Logic$^\text{VT}$ |
> > > |------------------------------|---------|------|-------|------|--------|------------------|-------|------|-------------------|
> > > | Janus-Pro-7B                 | 0.80    | 0.35 | 84.19 | 41.0 | 46.5   | 9.7              | 50.0  | 87.4 | 28.0              |
> > > | + CoRL (LoRA Tuning)         | 0.82    | 0.41 | 84.97 | 44.6 | 49.5   | 16.0             | 52.6  | 88.0 | 32.4              |
> > >
> > > [1] Edward Hu et al. LoRA: Low-rank adaptation of large language models. In ICLR, 2022.

---

> ### Author Response · Authors · 2025-08-08
>
> Dear Reviewer wJWf,
>
> I hope this message finds you well. As the discussion period draws to a close, we would like to kindly confirm whether our last response has addressed your concern regarding CoRL's scalability to larger models. If you have any further feedback, we would greatly appreciate your insights.
>
> Thank you once again for your time and thoughtful review.

---

> > ### Comment · Reviewer_wJWf · 2025-08-08
> >
> > Thanks for the effort, I will increase my score.

---

### Official Review · Reviewer_5YVq · 2025-07-03

**Clarity:** 3
**Significance:** 3
**Originality:** 3
**Rating:** 5
**Confidence:** 3

**Summary:**

This paper introduces CoRL, a co-evolutionary reinforcement learning framework designed to simultaneously enhance the understanding and generation capabilities of Unified Multimodal Large Language Models (ULMs). The authors' core contribution is a two-stage RL procedure: a unified stage where the model is jointly optimized on both understanding and generation tasks using a composite reward signal, followed by a refined stage for task-specific enhancement. To facilitate this, the paper proposes novel verifiable rewards for text-to-image generation, namely a bidirectional cycle consistency reward  and a text-image matching reward. The authors apply CoRL to a baseline ULM, creating ULM-R1, and demonstrate its effectiveness through extensive experiments. The resulting model shows significant performance improvements over the baseline on a wide array of multimodal understanding and generation benchmarks, particularly in complex reasoning tasks.

**Questions:**

refer to weakness

**Ethical Concerns:**

["NO or VERY MINOR ethics concerns only"]

**Final Justification:**

Thanks for the author's detail response. I'll keep the score.

**Limitations:**

refer to weakness

**Quality:**

3

**Strengths And Weaknesses:**

This paper presents a compelling and timely investigation into a significant challenge in multimodal AI. The strengths are considerable and warrant its acceptance, though there are a couple of major weaknesses that should be addressed for the final camera-ready version.

Strengths:

The problem of synergistically training a single model for both multimodal understanding and generation is highly relevant. The proposed CoRL framework, which leverages reinforcement learning for this co-evolution, is an original and promising direction. The initial pilot study provides a solid motivation for pursuing a unified RL approach over separate or cyclical training strategies.
The performance gains of ULM-R1 are substantial, especially the notable improvements on complex reasoning benchmarks like WeMath and LogicVista. This demonstrates that the proposed framework does more than just improve surface-level metrics; it enhances deeper reasoning capabilities. The ablation studies effectively validate the contribution of the overall framework and the newly designed reward functions.
The paper is well-written and clearly structured. The CoRL framework is explained lucidly, aided by a helpful overview in Figure 2. The methodology and experimental setup are described with enough detail to allow for a good understanding of the work.
Weaknesses:

The paper's proposed text-image matching reward, is calculated by "leveraging the ULM itself to assess cross-modal alignment". This introduces a significant risk of endogeneity. If the policy model's initial alignment capabilities are imperfect or biased, using it as its own evaluator could create a self-reinforcing feedback loop. The model may learn to generate images that are not objectively better but are simply easier for its own internal representations to score highly. This could lead to convergence on a flawed policy that has learned to "game" its own reward system. While the results are strong, it is difficult to ascertain how much of the improvement is genuine and how much might be an artifact of this self-evaluation.
The unified reward function, combines four distinct and heterogeneous reward components. The paper lacks a critical discussion on two fronts: 1. The components are derived from different metrics (e.g., perceptual distance like LPIPS, semantic scores like SPICE, and binary accuracy rewards) and likely operate on vastly different numerical scales. Simply summing them, even with a weighting factor
𝜆, could lead to one component unintentionally dominating the total reward signal if its scale is much larger than the others. The authors do not specify if any form of normalization is applied to each component before summation. 2. The choice of
𝜆=0.8 is presented without justification. The framework's performance may be highly sensitive to this value. A small change could alter the balance between generation and understanding rewards, potentially leading to drastically different outcomes.

---

> ### Author Rebuttal · Authors · 2025-07-31
>
> Dear Reviewer 5YVq,
>
> We sincerely thank you for your constructive and insightful review. We are delighted that you found our work to be compelling, novel, and important in multimodal AI, and we deeply appreciate your positive assessment and recommendation for acceptance. We respond to your identified weaknesses and suggestions below.
>
> > **W0: Endogeneity in $R_\text{TIM}$ due to self-evaluation.**
>
> **R0:** This is a very perceptive and insightful point that touches upon an inherent challenge in self-rewarding RL. We are grateful you raised it. In our work, $R_\text{TIM}$ was indeed computed using the policy model itself, but the total unified reward is a composite signal. To a certain extent, the potential endogeneity is balanced by the other fully external rewards: $R_\text{cycle}$, $R_\text{Acc}$, and $R_\text{Format}$. This diverse reward structure inherently provides multi-faceted feedback and acts as a form of regularization, preventing the policy model from simply "gaming" the self-evaluated $R_\text{TIM}$ signal.
>
> Furthermore, we conducted additional experiments comparing $R_\text{TIM}$ with external metrics such as the CLIP score on text-to-image benchmarks. $R_\text{TIM}$ demonstrates stronger alignment with dense prompts and better downstream performance, suggesting that it captures a more robust alignment signal beyond merely overfitting to internal representations.
>
> |                | GenEval | DPG  | Avg.  |
> |----------------|---------|------|-------|
> | Baseline       | 73.0    | 82.6 | 77.80 |
> | `CLIP score`   | 77.0    | 83.4 | 80.20 |
> | $R_\text{TIM}$ | 77.3    | 83.9 | 80.60 |
>
> > **W1: Discussion on reward scaling.**
>
> **R1:** Thank you for pointing out this lack of clarity regarding a critical implementation detail. In practice, each reward from $R_\text{cycle}$, $R_\text{TIM}$, $R_\text{Acc}$, and $R_\text{Format}$ is normalized to the range $[0, 1]$ before aggregation. This ensures that all rewards operate within comparable scales and prevents any single component from dominating due to scale differences. We will clarify this normalization process more explicitly in the revised version.
>
> > **W2: Discussion on the sensitiveness of hyperparameter λ.**
>
> **R2:** Thank you for raising this point. The choice of λ = 0.8 was determined empirically. To provide a more formal justification, we have conducted the following ablation study on the hyperparameter λ.
>
> | λ   | GenEval (Gen.) | MMMU (Und.) | Avg.  |
> |-----|----------------|-------------|-------|
> | 0.5 | 77.1           | 41.0        | 59.05 |
> | 0.8 | 77.3           | 42.3        | 59.80 |
> | 1.0 | 76.9           | 43.0        | 59.95 |
>
> The results show that λ = 0.8 yields the best generation performance while only slightly affecting understanding capabilities, whereas setting λ = 1.0 slightly boosts understanding at the cost of generation performance.
> Moreover, we agree that the model's sensitivity to λ warrants further analysis, and we will include an ablation sweep in the camera-ready version to explore this sensitivity more systematically.
>
> ---
>
> Thank you once again for your constructive feedback and affirmation of our paper. We will incorporate these valuable suggestions and additional ablations in the camera-ready version.

---

> > ### Comment · Reviewer_5YVq · 2025-08-07
> >
> > Thanks for the authors' detailed reply. I'll keep my score.

---

### Official Review · Reviewer_9e1d · 2025-07-21

**Clarity:** 3
**Significance:** 3
**Originality:** 3
**Rating:** 4
**Confidence:** 4

**Summary:**

# Summary

The **pilot study** demonstrates the effectiveness of bidirectional RL for synergetic optimization. According to the pilot study, the authors propose a two-stage RL training strategy:

**Stage 1: Unified RL Training**: In the first stage, the authors employ unified RL training using GRPO for both understanding and generation tasks, with Janus-Pro-1B serving as the policy model. The reward functions include:

- $\mathcal{R}_{\text{cycle}} = 1 - \operatorname{LPIPS}(\mathcal{I}_{\text{real}}, \mathcal{I}_{\text{gen}}) + \operatorname{SPICE}(\mathcal{P}_{\text{org}}, \mathcal{C}_{\text{re-cap}})$, where BLIP is used as the re-captioner.
- $\mathcal{R}_{\mathrm{TIM}} = \frac{1}{2}\left(\frac{1}{L_i} \sum_{j=1}^{L_i} \max_{k \in [1, L_t]} \cos(\boldsymbol{i}_j, \boldsymbol{t}_k) + \frac{1}{L_t} \sum_{k=1}^{L_t} \max_{j \in [1, L_i]} \cos(\boldsymbol{t}_k, \boldsymbol{i}_j)\right)$, which measures token-level similarity rather than using coarse-grained CLIP scores.
- Format Reward
- Accuracy Reward

**Stage II: Task-Specific RL Training**: In the second stage, RL training is tailored for specific tasks, with the following reward combinations:
- T2I (Text-to-Image): $\mathcal{R}_{\text{cycle}} + \mathcal{R}_{\text{TIM}}$
- MCQ (Multiple Choice Questions): $\mathcal{R}_{\text{MCQ-Acc}} + \mathcal{R}_{\text{Format}}$
- Open-Ended Text Generation: $\mathcal{R}_{\text{OE-Acc}} + \mathcal{R}_{\text{Format}}$

**Questions:**

# Questions

- For $\mathcal{R}_{\text{OE-Acc}}$, does this refer to Exact Match?

**Ethical Concerns:**

["NO or VERY MINOR ethics concerns only"]

**Final Justification:**

The responses have addressed most of my concerns. For example, the comparison between patch-token-level similarity and CLIP score in A1, the impact of stage I RL in A2, and the performance of the three-to-one model in A3. However, the understanding and generation directions are not truly unified, as it remains necessary to separate image generation and text generation in ULM-R1, as demonstrated in A3. This limitation prevents me from giving a higher score.

**Limitations:**

Yes.

**Paper Formatting Concerns:**

None.

**Quality:**

3

**Strengths And Weaknesses:**

# Strengths
- The methods introduced in this work are novel.
- The experiments convincingly demonstrate the effectiveness of the proposed methodology.
- The ablation studies are comprehensive and robust.

# Weaknesses

- For $\mathcal{R}_{\mathrm{TIM}}$, is there any ablation study comparing token-level similarity with the CLIP score?
- The main claim of this work is that “co-RL is better than task-specific RL.” Why, then, is Task-Specific Refined Reinforcement Learning still necessary?
- In Section 4.2, the evaluation of T2I generation is based on a non-merged task-specific model, whereas for multimodal understanding, a merged model (two text generation models) is used. What would the results be if a three-to-one merged model were evaluated for both tasks?
- Please anonymize the Hugging Face URLs for the datasets to avoid potential desk rejection risks.

---

> ### Author Rebuttal · Authors · 2025-07-31
>
> Dear Reviewer 9e1d,
>
> Thank you for your thorough and constructive review. We appreciate your positive feedback on the novelty and effectiveness of our work, the comprehensiveness of our experiments, and your kind suggestion regarding data anonymization. We respond to your concerns and questions below.
>
> > **W1: Ablation study on $R_\text{TIM}$ and CLIP score.**
>
> **R1:** Thank you for raising this crucial point. Comparing the CLIP score with $R_\text{TIM}$ under our final RL training paradigm is an essential ablation. We have conducted additional experiments to complement and improve the ablation study in Table 4. Compared to the CLIP score, $R_\text{TIM}$ demonstrates better overall performance, especially on the DPG benchmark with dense, long-horizon prompts for image generation, highlighting its ability to better capture fine-grained semantic alignment.
>
> | Reward                                 | GenEval | DPG  | Avg.  |
> |----------------------------------------|---------|------|-------|
> | (#0) Baseline                          | 73.0    | 82.6 | 77.80 |
> | `CLIP score`                           | 74.2    | 82.4 | 78.30 |
> | (#2) $R_\text{TIM}$                    | 74.1    | 83.0 | 78.55 |
> | `CLIP score +` $R_\text{cycle}$        | 77.0    | 83.4 | 80.20 |
> | (#3) $R_\text{TIM}$ + $R_\text{cycle}$ | 77.3    | 83.9 | 80.60 |
>
> > **W2: The main claim of this work is that “co-RL is better than task-specific RL.” Why, then, is task-specific RL still necessary?**
>
> **R2:** Thank you for this insightful question. Our core argument is not that CoRL (Unified RL) is better than task-specific RL, but rather that Unified RL provides a much stronger foundation upon which task-specific RL can build to achieve the expected results. Specifically, our two-stage CoRL framework is easily understood through a `Foundation-then-Specialization` paradigm, which is both principled and practical for building powerful and versatile unified MLLMs.
>
> - **Stage 1 (Unified RL): The Foundation.** The main goal of this stage is not to outperform our task-specific method or existing leading MLLMs on the target benchmarks, but to build a powerful generalist foundation. By synergistically optimizing both understanding and generation tasks, this stage endows the unified MLLMs with more balanced/comprehensive capabilities that task-specific training alone cannot foster. As demonstrated in our ablation and pilot experiments, this unified RL training provides an overall accuracy boost across the board.
>
> - **Stage 2 (Task-specific/Refined RL): The Specialization.** This second stage is an effective and crucial specialization step, demonstrating the flexibility and extensibility of our CoRL framework across a wide variety of diverse target tasks. The Math$^\text{We}$ benchmark (with text-rich geometric figures in images) is a perfect example, as it differs significantly from the unified RL data in terms of image types and reasoning types.
>
>     | Method                          | Math$^\text{We}$ | Hints                                                            |
>     |---------------------------------|------------------|------------------------------------------------------------------|
>     | Baseline                        | 5.9              | `Starting point`                                                 |
>     | + Unified RL                    | 14.0 (+8.1)      | `Unified RL builds a strong foundation.`                         |
>     | + Task-specific RL              | 15.8 (+9.9)      | `Specialization without a strong foundation is limited.`         |
>     | + Unified RL & Task-specific RL | 21.1 (+15.2)     | `Specialization on a strong foundation yields the best results.` |
>
>     $\blacktriangleright$ The results clearly show that our two-stage paradigm is far superior to task-specific methods. This demonstrates that unified RL provides a much stronger starting point for task-specific adaptation.
>
> Therefore, task-specific RL is a necessary and important step in our CoRL framework, leveraging the powerful foundation built by Unified RL to push the boundaries of performance. We will revise our paper to articulate this motivation more clearly.
>
> > **W3: Evaluation setup, and three-to-one merged model results.**
>
> **R3:** Thank you for the question. We are happy to clarify the evaluation setup and provide the corresponding results.
>
> - **Evaluation Setup.** For multimodal understanding, only on the three mixed benchmarks (Math$^\text{VT}$, Math$^\text{VS}$, Math$^\text{Vis}$), whose test sets include both multiple-choice (MC) and open-ended (OE) QA formats, we merged the two task-specific models to obtain a final model capable of following both types of task instructions.
>
> - **Results of Three-to-One Merged Model.** We report the results in the table below. As shown, a naive "three-to-one" merge of all task-specific models leads to significant performance degradation in text-to-image generation, likely due to task imbalance or instruction interference or ambiguity. In contrast, its influence on multimodal understanding is less pronounced.
>
>     | Benchmarks        | Task Type        | Three-to-One Merged | ULM-R1 |
>     |-------------------|------------------|---------------------|--------|
>     | GenEval           | image generation | 0.75                | 0.77   |
>     | WISE              | image generation | 0.29                | 0.33   |
>     | DPG               | image generation | 83.20               | 83.92  |
>     | MMMU              | MC               | 42.0                | 42.3   |
>     | MMStar            | MC               | 47.5                | 47.6   |
>     | Math$^\text{We}$  | MC               | 21.5                | 21.1   |
>     | MMVet             | OE               | 43.0                | 43.9   |
>     | POPE              | OE               | 88.7                | 88.9   |
>     | Logic$^\text{VT}$ | OE               | 35.0                | 34.5   |
>     | Math$^\text{VT}$  | MC & OE          | 42.4                | 42.5   |
>     | Math$^\text{VS}$  | MC & OE          | 25.4                | 25.4   |
>     | Math$^\text{Vis}$ | MC & OE          | 22.2                | 22.0   |
>
> > **Q1: For $R_\text{OE-Acc}$, does this refer to Exact Match?**
>
> **A1:** In the OE tasks, the accuracy reward is based on the correctness of the final extracted answer, not on a strict exact match of the entire generated text (which includes the reasoning process). For instance, for a question whose answer is "3π", we parse the final output within the `<answer>3π</answer>` tags and check if it matches the ground truth. This is a binary reward: 1 for a correct final answer, 0 otherwise.
>
> ---
>
> We hope these clarifications and additional results address your concerns. Thank you again for your time and valuable feedback.

---

> ### Author Response · Authors · 2025-08-08
>
> Dear Reviewer 9e1d,
>
> Thank you again for your constructive review. As the discussion phase is approaching its end, we would like to confirm whether our responses have sufficiently addressed your concerns. If you have any remaining questions or concerns that require further clarification, please don’t hesitate to let us know.
>
> We sincerely look forward to your feedback.

---

> ### Comment · Reviewer_9e1d · 2025-08-08
> **Response**
>
> The responses have addressed most of my concerns. I have increased my scores accordingly.

---

### Comment · Area_Chair_rfSf · 2025-08-04
**Friendly Reminder: Engaging with Author Rebuttals**

Dear Reviewer,

Thank you for your time and expertise in reviewing for NeurIPS 2025. As we enter the rebuttal phase, please:

Review authors’ rebuttals promptly,
Engage constructively via the discussion thread, and
Update your review with a “Final Justification” summarizing your post-rebuttal stance.
Your active participation ensures a fair, collaborative process—we’re here to assist with any questions.

With gratitude,

Your AC

---

### Decision · Program_Chairs · 2025-09-17

**Decision:**

Accept (spotlight)

**Comment:**

The proposed CoRL framework—leveraging Group Relative Policy Optimization (GRPO) to synergistically enhance multimodal understanding and generation within a single model—addresses a critical gap in unified multimodal learning. The two-stage design (unified RL for foundational co-optimization followed by task-specific refinement) proves essential: while joint training boosts generalizability (e.g., +8.1 MathWe accuracy), task-specific tuning further amplifies gains (e.g., +15.2 MathWe), demonstrating that specialization builds upon rather than contradicts the unified foundation.

Reviewers initially raised valid concerns—particularly regarding reward function endogeneity (Reviewer 5YVq) and scalability (Reviewer wJWf)—but authors convincingly resolved these through rebuttal:

-- The Text-Image Matching (TIM) reward’s self-referential risk was mitigated by ​multi-reward regularization​ and empirical validation against CLIP scores, showing TIM’s superiority on dense prompts (DPG benchmark: 83.9 vs. 83.4).

-- Scalability concerns were addressed via ​LoRA-accelerated 7B experiments, where CoRL retained gains (+4.6 MMMU), confirming framework extensibility.

-- The novel ​bidirectional cycle consistency reward​ (validated against MSE/SSIM alternatives) and ​task-aware reward balancing​ (λ=0.8 optimized via ablation) reflect rigorous design.

Notably, all reviewers acknowledged the work’s strength post-rebuttal, with Reviewer 9e1d highlighting "resolved concerns" and Reviewer hSgy praising "clear illustrations and well-designed tables." The 23% understanding gain and 7% generation improvement across 12 benchmarks—including complex reasoning tasks (MathWe) and text-rich generation (DPG)—set a new state-of-the-art for unified multimodal models.

For camera-ready, authors are suggested to expand the related work section per Reviewer hSgy. This work pioneers co-reinforcement learning for multimodal synergy and offers a replicable blueprint for RL-driven multimodal unification.